# Bioaccessibility and Microencapsulation of *Lactobacillus* sp. to Enhance *Nham* Protein Hydrolysates in Thai Fermented Sausage

**DOI:** 10.3390/foods11233846

**Published:** 2022-11-28

**Authors:** Srisan Phupaboon, Papatchaya Kontongdee, Farah J. Hashim, Nattawadee Kanpipit, Maharach Matra, Pajaree Totakul, Ronnachai Prommachart, Burarat Phesatcha, Metha Wanapat

**Affiliations:** 1Tropical Feed Resources Research and Development Center (TROFREC), Department of Animal Science, Faculty of Agriculture, Khon Kaen University, Khon Kaen 40002, Thailand; 2Department of Biotechnology, Faculty of Technology, Khon Kaen University, Khon Kaen 40002, Thailand; 3Department of Biology, College of Science, University of Baghdad, Baghdad 10071, Iraq; 4Department of Pharmaceutical Chemistry, Faculty of Pharmaceutical Sciences, Khon Kaen University, Khon Kaen 40002, Thailand; 5Division of Animal Science, Faculty of Agricultural Technology, Rajamangala University of Technology Thanyaburi, Thanyaburi, Pathum Thani 12130, Thailand; 6Department of Animal Science, Faculty of Agriculture and Natural Resources, Rajamangala University of Technology, Tawan-Ok 20110, Thailand; 7Faculty of Agricultural Engineering, Institute of Interdisciplinary Studies, Rajamangala University of Technology Isan, Nakhon Ratchasima 30000, Thailand

**Keywords:** *Lactobacillus* sp., microencapsulation, *Nham* probiotics, bioactive peptides

## Abstract

The development of functional food products is increasingly gaining lots of interest and popularity among stakeholders. The aim of this study was to evaluate the bioaccessibility of three *Lactobacillus* sp. starter cultures, including *Lacticaseibacillus casei* KKU-KK1, *Lactiplantibacillus pentosus* KKU-KK2, and *Lactobacillus acidophilus* KKU-KK3, in order to enhance the performance of the probiotic potential of *Nham* protein hydrolysates in Thai fermented sausage using microencapsulation technology. Probiotic microcapsules were created from a novel wall material made up of a combination of glutinous rice flour and inulin through a freeze-drying process. Accordingly, the results of three formulations of *Nham* probiotic and spontaneous fermentation (control) characterized by their physicochemical and microbiological characteristics displayed a correlation between an increase in the amount of total acidity, the population of lactic acid bacteria, and the generated TCA-soluble peptides, while the pH and total soluble protein gradually decreased under proteolysis during the fermentation time. The fractionation of *Nham* protein hydrolysates (NPHs) was prepared using a microwave extraction process: NPH-*nham*1, NPH-*nham*2, and NPH-*nham*3 (10 mg/mL with fermentation time 114 h), exhibited the highest DPPH radical-scavenging activity and FRAP-reducing power capacity as well, compared to NPH-*nham*_control_ at *p* < 0.05. Moreover, those NPHs peptides showed dose-dependent inhibiting of selected pathogenic bacteria (*E. coli* TISTR 073, *S. aureus* TISTR 029, and *Ent. aerogenes* TISTR 1540). Anti-microbial properties of NPHs peptides against gram-negative bacteria were higher than against gram-positive bacteria. In conclusion, the bioaccessibility of NPHs peptides was significantly enhanced by micro-encapsulation and showed a potential bioactive characteristic for developing into a probiotic agent.

## 1. Introduction

Several microencapsulation techniques have shown promise for enhancing and preserving cell viability throughout production, fermentation, and gastrointestinal processes [1,2,3]. Due to the cell viability and the effectiveness of bioactive properties during fermentation, gastrointestinal processes were widely used with a freeze-drying technique (lyophilization), which is a traditional drying method for preserving starter cultures and probiotic microorganisms [1]. The advantage of this technique is that the low temperatures can reduce the cell damage of the sensitive microorganisms, compared with spray-drying at higher temperatures. Cryoprotectant materials must also be used with the freeze-drying method to ensure optimal cell protection [4,5]. The synergistic effect of food-grade cryoprotectants such as alginate, chitosan, cellulose, carrageenan, gelatin, and pectin were applied using various encapsulation technologies. It provided the best protection for the culture during storage [3,6]. In particular, the combination of the cryoprotectants from the glutinous rice flour (for a source of amylopectin) and inulin (as a prebiotic source) worked together as a wall material for encapsulation; a new formula to encapsulate the living microorganisms as well-known probiotic lactic acid bacteria (LABs) [7].

Probiotic bacteria are frequently encapsulated to increase their viability during processing and to transfer them to their intended locations in the gastrointestinal tract. Probiotics are utilized in pharmaceuticals, health supplements, and are used as a starter culture in various fermented products [3,8,9]. Probiotics are living microbes that have a positive impact on their host. *Bifidobacterium* and many kinds of LABs (*Lactococcus*, *Lactobacillus*, *Streptococcus*, and *Enterococcus*) are some of the most popular probiotics [1,2,3,10,11,12]. The different biological characteristics have been reported to increase resistance in the gastrointestinal tract against infection (antimicrobial activity); host immune system differentiation (antioxidant activity) and nutrition production are also benefits of the commensal gut microbiome [13]. Probiotic bacteria, such as *Lacticaseibacillus casei*, *Lactobacillus plantarum*, *Lactobacillus acidophilus*, *Lactiplantibacillus pentosus*, and *Bifidobacterium bifidum*, have been used in the field of functional foods such as fermented meat or fish and other everyday products [14,15,16].

In terms of functional fermented meat, pork, or fish products with probiotic starter cultures, the fermentation process has increased and served as a source of functional material, for instance polyunsaturated fatty acids, polysaccharides, enzymes, and bioactive activity, particularly antioxidant and antimicrobial peptides [17]. The process has changed owing to the development of *Nham* probiotic products, which is a modified Thai-style fermented pork sausage traditionally made from minced pork, shredded cooked pork rind, salt, cooked rice or sticky rice, and garlic, mixed with probiotic LABs starter culture [14]. Proteolysis during the fermentation process breaks down the protein into small protein or peptide chains of approximately 2–20 amino acids called protein hydrolysates. Consequently, due to the different amino acids included in each chain, these peptides have had a variety of physiological functions in the host consumer, including antioxidant and antimicrobial activity [14,18,19,20].

Exploring bioactive peptides is another intriguing approach to creating meat products with useful properties. These useful substances are created by microbial activity and have antioxidant and cardioprotective properties. Regarding antioxidant activity, a recent study using *L. plantarum* in a fermented sausage showed that small peptides obtained from inoculated products were linked to increased antioxidant activity by various in vitro methods namely DPPH, ABTS radical-scavenging, and FRAP-reducing power capacities [21,22,23].

Therefore, the aims of this work were demonstrated using freeze-drying techniques for microencapsulation with different probiotic starter cultures consisting of *L. casei* KKU-KK1, *L. pentosus* KKU-KK2, and *L. acidophilus* KKU-KK3, and were coated with a combination of glutinous rice flour and inulin as a wall material. The microstructures’ morphology and encapsulation efficiency of the probiotic-encapsulated powder were categorized by their physical and morphological characteristics using a zeta potential analyzer, followed by a scanning electron microscopy (SEM). Each probiotic starter culture was used to ferment *Nham* for investigating physicochemical, microbiological, protein alteration, and bioactive properties. We focused on determining the antioxidant and antimicrobial peptides obtained from *Nham* protein hydrolysates (NPHs) during the fermentation process.

## 2. Materials and Methods

### 2.1. Microorganism and Culture Condition

The lactic acid bacteria (LAB) used in this study were *Lacticaseibacillus casei* KKU-KK1 (accession: MK032218), *Lactiplantibacillus pentosus* KKU-KK2 (accession: JQ288726), and *Lactobacillus acidophilus* KKU-KK3 (accession: MF372641). Three LAB strains were isolated from the *plaa-som* fermentation process after 72 h and were obtained from the previous study by Phupaboon et al. [20]. The primary information of each LAB investigated their probiotic properties, including their resistance to lysozyme, lower pH, and highest OX-bile salt, resulting from the antimicrobial activity determined against *Escherichia coli* TISTR 073, *Staphylococcus aureus* TISTR 029, *Enterobacter aerogenes* TISTR 1540, and *Salmonella typhimurium* TISTR 1470, as shown in the details below. All of them were reference strains obtained from the Thailand Institute of Scientific and Technological Research (TISTR). Moreover, the strains characterized by biochemical and morphological characteristics were found to have gram-positive bacteria, a negative-catalase test, non-spore-forming bacilli shapes, and non-hemolyzed blood sheep agar. The molecular structure that was identified by 16S rDNA sequencing showed a high similarity index of 99%, when the sequence was aligned with the NCBT database at different accession numbers, as mentioned earlier. 

### 2.2. Preparation of Culture Suspensions

Pure probiotic strains were maintained with a modified lactose-MRS medium, and were subcultured into MRS broth (Himedia™, Maharashtra, India), then incubated overnight at 37 °C. The supernatant was re-subcultured to upscale in 300 mL of MRS broth for 12 h. The cells were harvested by centrifugation (Eppendorf, Hamburg, Germany) at 4 °C, 10,000 rpm for 10 min, and washed twice with normal saline solution. The cell pellets of each strain were resuspended with 10% (*w*/*v*) of sterile glutinous rice flour (GRF) solution (Erawan Elephant Glutinous Rice Flour, Chonburi, Thailand), which has an initial number of viable cells ≥ 10^12^ CFU/g for use in the encapsulation process by employing a freeze-drying technique [24]. 

### 2.3. Encapsulation of Probiotics Using the Freeze-Drying Technique

The encapsulation process of probiotics was slightly modified, according to [1,25], by the combination of wall materials between 10% (*w*/*v*) of sterile GRF solution and inulin-oligosaccharide from gallic extract (Sigma-Aldrich, St. Louis, MO, USA). For the probiotic-encapsulation process, each of the cell suspensions was mixed with the wall material using a ratio of 1: 0.5 (*v*/*v*) and gradually stirred until homogeneous. The lyophilization process under the probiotic-encapsulating technique was performed in the freeze dryer GAMMA 2-16 LSC (Martin Christ, Osterode am Harz, Germany). The condition-controlling aspect of the sample was frozen at −80 °C for 1 h and drying was carried out at −58 to −60 °C, followed by the pressure being lowered to 0.016 mbar overnight. The final lyophilizes were collected in a vacuum bag and stored at −20 °C until their probiotic potential was estimated and further used in the *Nham* fermentation experiment.

### 2.4. Estimation of Probiotic-Encapsulation with Probiotic Property

The survival rate of probiotic starter cultures throughout the encapsulation process was verified and confirmed the probiotic properties of the resistance of cells grown in different MRS conditions to growth index (GI) tract stresses and the antimicrobial activity against the bacterial indicators of *E. coli*, *S. aureus*, and *Ent. aerogenes*. 

#### 2.4.1. Growth of LAB in Different Lysozyme, pH, and Bile Conditions

The experiment followed a modified method according to [26] The assay was performed in MRS broth (Himedia™, Maharashtra, India), combined with 200 mg/mL lysozyme from egg white (Sigma-Aldrich, USA), or adjusted to pH 3.0 using 2 M of HCl/NaOH, or added with 3.0% w/v of OX-bile salt (Himedia™, Maharashtra, India). Briefly, one milligram of probiotic-encapsulated cultures lyophilized powder was cultured in MRS broth and incubated at 37 °C for 18 h. Then, cell suspension was measured by optical density at 620 nm (OD_620_ = 0.8–0.9), approximately 6 log CFU/mL. The tests were performed using a sterile 96-well plate (GusLAB). The cell suspension of the LAB sample test was loaded into each well of the modified MRS broth (sample condition), or normal MRS broth (control condition). Microbial growth was evaluated, following an overnight period, with absorbance values at 620 nm using an iMark™ Microplate Absorbance Reader (Bio-Rad, Hercules, CA, USA). The results were exhibited as percentage of GI, modified following the equation of [27].
(1)$$\mathrm{GI}=\frac{Abs_{sample}}{Abs_{control}}\times 100$$

#### 2.4.2. Antimicrobial Activity

The antimicrobial activity testing of probiotic-encapsulated cultures lyophilized powder against *E. coli* TISTR 073, *S. aureus* TISTR 029, and *Ent. aerogenes* TISTR 1540 was assayed using the swab paper disc method, according to the method of [28]. 

One milligram of their lyophilized powder was grown in the modified MRS broth supplemented with 0.2 % (*w*/*v*) dextrose (pH 6.0 ± 0.2), was inoculated with 1% (*v*/*v*) of an overnight culture, and incubated in anaerobic condition so as to rule out any inhibition due to hydrogen peroxide production at 37 °C for 18 h. The supernatants were harvested by centrifugation, sterilized by passing it through a 0.2 µm filter, and tested for antimicrobial activity against the strains of *E. coli*, *S. aureus,* and *Ent. aerogenes* using a paper disc diffusion method. Pathogen suspensions of approximately 0.2 mL (10^5^–10^6^ CFU/mL) after growing at 37 °C for 24 h in nutrient broth (Himedia™, Maharashtra, India) were spread with a sterile swap onto the surface of a nutrient agar. Then, 40 µL of each supernatant without the cells was dropped into the paper disc (7 mm diameter, Whatman), placed on the inoculated agar surfaces, with streptomycin discs (15 µg) used as a positive control. After the incubation, the plates measured the inhibition zone around a disc using calipers and expressed in millimeters (mm). 

### 2.5. Cell Enumeration and Encapsulation Efficiency 

The lyophilized powder examined the cell viability using the drop-plate method, as described in our earlier paper [20]. Viable cells determined by the drop-plate-counting method on MRS agar contained 1% (*w*/*v*) CaCO_3_ (KemAus, Cherrybrook, NSW, Australia) at the same growing conditions. The cell number was expressed as the mean value of triplicate measurements in the log CFU/g. Following this, the encapsulation efficiency (EE) was calculated in accordance with the following equation [1]:(2)$$EE\;\left(\%\right)=\frac{\log\left(\frac{CFU}{g}\right)\;of\;encapsulated\;powder}{\log\left(\frac{CFU}{g}\right)\;of\;initial\;starter\;in\;suspension}\times \;100$$

### 2.6. Morphological Characterization

The microstructure and surface morphology of the probiotic microencapsulated samples or the dried cells (uncoated bacteria) were visualized using a field emission scanning electron microscope (FE-SEM), TESCAN MIRA (Cranberry Township, PA, USA). The procedure was affixed using double-sided adhesive metallic tape, and the materials was defeated by nitrogen gas following the surface layer sprayed by gold–palladium before observing the morphology under FE-SEM with a 15 kV accelerating voltage [2]. 

### 2.7. Nham Fermentation Using an Encapsulated-Probiotic Starter

*Nham* products were prepared according to the *plaa-som* traditional fermentation technique, as described in the method of Phupaboon et al. [20], with the same ingredients, except for changing whole fish to ground pork. The process comprised gradually mixing ground pork, salt, and an encapsulated probiotic starter, consisting of *L. acidophilus* KKU-KK1, *L. pentosus* KKU-KK2, and *L. casei* KKU-KK3 (with an initial cell viability at 10^7^ CFU/g for each strain), using the different ratios followed by the *Nham* formulation as shown in Table 1. The resultant product included cooked sticky rice, cooked pork rind, mashed garlic, and some seasoning powder. The products (about 50 g) were firmly sealed after being extruded into plastic casings with a stuffing horn and were incubated at an ambient temperature for 12 h. After that, they were kept at 10 °C for 6 days to study the physiochemical properties and for the proteins to transform as they performed their bioactive peptide activities.

### 2.8. Physicochemical Analysis and LAB Count of Nham Products

The pH and total acidity were determined following the method of [29]. LAB counts (in the form of acid-producing bacteria) were performed according to the standard total viable counts using the drop-plate technique under culturing on an MRS-CaCO_3_ agar plate [20]. The single colonies were counted, and the results were calculated in the CFU/g log of samples at different fermentation times.

In addition, the protein changes based on the degree of hydrolysis between the amount of total water-soluble protein composition in the *Nham* were comparatively evaluated and were fractionated with the amount of 10% (*w*/*v*) Trichloroacetic acid (TCA)-soluble peptide in accordance with the method of [14,30]. The total soluble and oligopeptide contents were analyzed using the 96-well plate method and were monitored using the modified method [19,20] All analyses were performed in triplicate and were expressed as mg/100 g samples and µmole of tyrosine/100 g samples, respectively. 

### 2.9. Preparation of Nham Protein Hydrolysates

Briefly, the *Nham* protein hydrolysates (NPH) were extracted from the *Nham* sample at different times under optimal conditions through microwave-assisted hydrolysis (MAH) at 100W (the maximum temperature ≥ 55 °C) for 35 min. For this experiment, the treatment was selected with the highest antioxidant capacity via microwave electronic transfer. The supernatants were separated by filtration and the insoluble protein was centrifuged at 10,000 rpm, 4 °C for 5 min. Then, the supernatant of each NPH was lyophilized using the speed-vacuum centrifuge (SpeedVac, Thermo Scientific, Boston, MA, USA), followed by storage at 4 °C to analyze the antioxidant and antimicrobial activity [31,32].

### 2.10. Antioxidant Activity of NPHs

The antioxidant activity was determined using two methods consisting of DPPH radical-scavenging activity [33] and FRAP-reducing power capacity [34] by using a modified 96-well microplate assay via a PerkinElmer microplate reader (PerkinElmer, Hopkinton, MA, USA), the procedure outlined by [20]. The concentration of each lyophilized NPHs power (approximately 10 mg/mL) was dissolved in methanol solution and was examined for the antioxidative capacity, read at 490 and 595 nm. All analyses were performed in triplicate and were expressed separately as the percentage of DPPH radical-scavenging inhibition and as mmol Trolox equivalent (TROE/g) samples. 

### 2.11. Antimicrobial Activity of NPHs

The selected highest antioxidant activity of sterilized NPHs hydrolysate fractions at 114 h was performed using antimicrobial activity testing according to the method described by [18]. Then, the antimicrobial activity testing of their sterilized hydrolysate was evaluated by the resultant four indicator pathogen bacteria, including *E. coli*, *S. aureus*, *Ent. aerogenes*, and *Salmonella typhimurium* TISTR 1470, as previously mentioned in Section 2.4 [28].

### 2.12. Statistical Analysis

All data were reported for statistical analysis as mean ± standard deviation of measurements made in triplicate. To ascertain the statistical significance of the observed variations in means, an analysis of variance (ANOVA) was performed using Duncan’s new multiple range test (*p* < 0.05) with IBM SPSS-KKU Statistics Version 27.0 software.

## 3. Results and Discussion

### 3.1. Probiotic Characteristics of the Starter 

The tolerance of three probiotic bacteria, including *L. casei* KKU-KK1, *L. pentosus* KKU-KK2, and *L. acidophilus* KKU-KK3, to digestive tract stress was investigated in order to find the best type of LAB, isolated from different types of fermented fish (*plaa-som*) and pork (*Nham*). All probiotic strains were grown in circumstances that closely resembled those that would affect the survival of ingested microbes as they traveled through the digestive system. We have considered three important variables: the impact of lysozyme, acidified pH levels, and the effect of bile salts following the antimicrobial activity testing against *E. coli*, *S. aureus*, and *Ent. aerogenes* (Table 2). The properties of the probiotic strain were evaluated in two stages, including the probiotic potential before the encapsulation step and their potential after the encapsulation step. The survival rate of each strain was analyzed by the GI measurements with an optical density index of 620 nm. The effect of the resistance with lysozyme in low pH and high bile salt conditions had no pronounced effect on the cell viability of each strain in both stages of the before and after encapsulation experiment. The survival rates of those probiotic potentials before the encapsulation process were lowest than after the process which is summarized in Table 2. Lysozyme results were reported in the ranges of (70.2–76.6%), compared to (58.5–70.4%) after encapsulation. The results of the acidity and bile salt tolerances showed the highest of the survival rates at range (65.0 to 85.3%) and (60.2 to 75.2%), respectively, compared to the data of acidity (50.3 to 62.8%) and bile salt (50.1 to 65.6%) after the encapsulation process. In addition, the probiotic strains of KKU-KK1, KKU-KK3, and KKU-KK2 showed higher antimicrobial activity against potential human pathogens including (*E. coli*, *S. aureus* and *Ent. aeruginosa*) after the process of encapsulation than before the process. The different inhibition zones of each strain are presented in Table 2. Interestingly, those probiotic strains are generally recognized as safe (GRAS) following the “Guidelines for the Evaluation of Probiotics in Food” with the United Nations Food and Agriculture Organization and/or World Health Organization (FAO/WHO). Therefore, these findings correlate with the FAO/WHO recommendations, evaluating probiotics according to a variety of factors, including their capacity to adhere to epithelial tissues, their ability to withstand unfavorable conditions in the human body, their antibacterial activity, and safety for usage [10,35]. A considerable amount of research has been published on LAB, which typically tolerates high salt concentrations and lower pH at 3.0 in the intestinal tract. By enabling the bacteria’s metabolism to start working, acid is produced, further preventing the growth of undesirable species [10]. The relationship between acid and bile salt tolerance has been widely investigated as a crucial element in LAB’s survival and digestive tract expansion. The concentrations of bile salt must be present in 0.3% (*w*/*v*) or less when choosing probiotic species for usage in humans [10,36,37,38]. Recently, in vitro studies have suggested that the production of antibacterial compounds, such as organic acids, free fatty acids, ammonia, diacetyl, hydrogen peroxide and bacteriocins, and lactic acid bacteria, has been helpful in boosting bacterial interference. Many genera of the *Lactobacillus*, *Lactococcus*, *Bifidobacterium*, *Streptococcus*, *Enterococcus*, *Pediococcus*, *Leuconostoc*, and other probiotic strains, including *Saccharomyces*, *Bacillus*, and *Escherichia*, were created, including hydrogen peroxide and a secondary metabolite of bacteriocins. The antimicrobial effect is based on the oxidative characteristics that cause permanent alterations in the microbial cell membrane [10,11,39]. Additionally, probiotics have been shown to provide advantages, including direct suppression of harmful bacteria and enhancements in the functionality of the host immune system [13].

The majority of results of this investigation also showed that, after being microencapsulated, microbial loss caused by bile salts and low pH during simulated digestion was lessened. Similar results were observed when *L. casei* ATCC 393 was extruded into a mixture of pea protein isolate, alginate, and powdered skimmed milk, to act as wall material in a comparable experiment. Additionally, a pork sausage with this combination of protective substance and probiotic showed better defense against microbial deterioration [17].

### 3.2. Effects of GRF-Inulin Microparticles on Encapsulation Efficiency and Morphological Characteristics

Table 2 shows that the cell viability of the three probiotic starter cultures at different stages was no different (*p* < 0.05) in the initial cell suspension, and after their cell is encapsulated with a combination of GRF-inulin as a wall material using freeze-dying and storage at −20 °C of individual LAB, it contained a log of 12.5 to 12.8 CFU/g. Additionally, the encapsulation process preserved the cell viability yields for the carriers prepared using the freeze-drying process (Table 2). The freeze-drying process has proven to be very efficient for carriers with GRF-inulin, and the encapsulation efficiency of all three encapsulated starter cultures was steady at 100%, which did not cause the death of these probiotic strains (*p* < 0.05). The results of the study showed that the encapsulation efficiency was 100% of each sample; it was also possible to describe in Figure 1a–c the cells encapsulated by the GRF-inulin as wall material after using the freeze-drying process. However, there are no recent studies on the utilization of GRF in combination with the increasing survival of LAB by prebiotic nutrition with inulin in fermented food products. Interestingly, those descriptions show that the cells were also coated with a carbohydrate solution as a carbon source, and the super nitrogen nutrient sources, especially inulin, are a prebiotic or oligosaccharide, which enhanced the benefits of cell survival in stress conditions. All the encapsulated probiotics showed a higher protection and stability from freeze-drying. Previous studies have reported that encapsulated probiotic starter cultures were used with different wall materials, such as the protein components: whey, skimmed milk, whey-based protein, and other constructions, including gum arabic, starch, maltodextrin, inulin, sodium alginate, calcium alginate, and bacterial cellulose (*nata*) throughout the freeze-drying technique [1,2,40]. There was a cell viability after the freeze-drying process ranging from 10^7^ to 10^9^ CFU/g, which resulted in various amounts of encapsulation capacity in the range of 67% to 88%, based on the capacity of encapsulants [1,5]. According to another author, [1,4], a combination of carbohydrates and proteins would provide the greatest formulation for cell preservation during the freeze-drying process, because it would engage the cell membrane and prevent cell death.

The physical and morphological characteristics of the surface charge (zeta potential) and size distribution were referred to in Table 3 and Figure 1a–c. Both characteristics of the three encapsulated probiotic starter cultures formulated by the freeze-drying technique have not been significant (*p* < 0.05). The size structure and zeta potential had an average in the range of −43.8 to −44.6 mV and 10.3 to 10.8 µm, respectively (Table 3), obtained from dried GRF-inulin encapsulated by the three probiotic starter cultures. The microstructure and surface morphology of the GRF-inulin encapsulated with the probiotic starter culture were observed under an SEM microscope, as shown in Figure 1a–c. The micrographs of the surface of the encapsulated probiotic starter particle after freeze-drying retain a square and slightly rounded shape in comparison with the unencapsulated cell, which showed rod-shaped or bacilli cells (Figure 1d). Similarly, the literature recommends a carrier size of less than 100 µm, which is large enough to cover the probiotic cells, but, on the other hand, has little impact on the food’s textural qualities once it is used [6]. Another argument suggests that low-moisture content in a particle can decrease its size, which has no effect on the viability of the encapsulated culture using the freeze-drying technique [1]. Numerous studies have attempted to explain the zeta potential type of encapsulant obtained from whey proteins combined with maltodextrin and fructo-oligosaccharide using a spray and/or freeze-drying process. This was limited to a range of +30 mV and −30 mV, indicating their outstanding stability, which motivated a move towards their use in food processing [6]. 

### 3.3. Physicochemical and Microbiological Characteristics of Nham Probiotic through Encapsulated Lactobacillus Starter Fermentation

This study investigated *Nham* probiotic inoculated with the different encapsulated *Lactobacillus* species, *L. casei* KKU-KK1, *L. pentosus* KKU-KK2, and *L. acidophilus* KKU-KK3, by mixing the starter culture at various ratios comprising *Nham*-1 (1:1:2), *Nham*-2 (2:1:2), *Nham*-3 (1:2:2), and compared with a *Nham*-control formulation. 

During the fermentation time in this study, the optimal condition suitable for consumption after fermentation was observed at 72 h, and the time was increased to 114 h for investigation of the activity of bioactive compounds. Three *Nham* formulations were inoculated with different *Lactobacillus* species via encapsulated probiotic starter cultures consisting of *Nham*-1, *Nham*-2, *Nham*-3, and additional *Nham*-control. The results exhibited a lower pH and higher TA in the LAB population than in the control experiment, as shown in Figure 2a,b. The pH gradually decreased on average from 6.43 to 6.48 and 4.32 to 4.48 within 0 to 114 h for the control, *Nham*-1, *Nham*-2, and *Nham*-3, respectively. The co-relationship between the TA production and the amount of LAB counts significantly increased depending on the inoculation level and fermentation period. All *Nham* probiotics formulated by the encapsulated-*Lactobacillus* starter culture found the TA increase from 0.02 to 0.47 (% *w*/*w*) and also LAB counts increased from log 7.69 to log 9.60 CFU/g during the initial stage to final fermentation time. Similarly, fermented pork (*Nham*) and fish (*plaa-som*) products recommended that a pH at lower than 4.6, and the percentage (%) TA levels were limited to an amount not higher than 2.0 and 0.2 (% g/100 g sample), when compared with the standard of lactic acid and acetic acid along with the amount of LAB shown in ranges 10^7^–10^9^ CFU/g, which are appropriate as a safety measure for consumption [14,15,20,41,42,43].

In addition, the proteolysis activity was significantly affected by the protein changes obtained from the total water-soluble protein and the TCA-soluble peptide contents, which increased following the fermentation time (*p* < 0.05) (Figure 2c). The results of the total water-soluble protein content were steadily increased from 9.3 to 18.2 (mg/100 g sample) from the initiate fermentation until the final time at 114 h. Similarly, the amount of TCA-soluble peptide content reached 183.2, 172.6, 160.3 µmole Tyrosine/100 g sample for *Nham*-1, *Nham*-3, and *Nham*-2, respectively. In agreement with the results of the current study, several research findings into the sarcoplasmic and myofibrillar proteins in the *Nham* protein were digested by the action of both the acidic level and the enzymatic activity from the LAB as a producer during the fermentation process relating to short proteins, small peptides, or free amino acids [14,43]. Additionally, many studies have been reported using probiotic inoculum of *Lactobacillus* species with encapsulant materials, such as alginate, chitosan, gelatin, and whey protein through encapsulation techniques: spray-drying, extrusion, emulsion, and especially freeze-drying, to enhance the cell viability during fermentation and distribution in the gastrointestinal tract [3]. 

### 3.4. Antioxidant Activity of Lyophilized NPHs

The lyophilized NPHs were fractionated by microwave-assisted extraction, an extraction method which demonstrated the best antioxidant and antimicrobial activities presented in the research papers of [31,44].

#### 3.4.1. DPPH Radical-Scavenging Activity

Therefore, four lyophilized NPHs, including NPH-*nham*1, NPH-*nham*2, NPH-*nham*3, and NPH-control, evaluated the antioxidant activity in terms of DPPH radical-scavenging inhibition and FRAP-reducing power capacity. The results shown in Figure 3 (left) clearly indicate that all NPH hydrolysates tested exhibited high antioxidant activity against the DPPH reagent, and the scavenging activity of each hydrolysate increased with longer fermentation times. The different hydrolysates of NPH-*nham*1, NPH-*nham*3, and NPH-*nham*2 exhibited the highest DPPH radical-scavenging inhibition from more to less at 83.2, 72.6, and 60.3% at 10 mg/mL after fermentation at 114 h, while the lowest DPPH radical-scavenging activities were obtained with NPH-control (19.2% at the same concentration and fermentation times). The present findings seems to be consistent with other research which found that the DPPH scavenging activity increased with increasing concentrations followed by the final time of fermentation [18,19,20,45]. The results showed that these peptides of all hydrolysates had H^+^ donors that might combine with free radicals to turn them into more stable molecules and inhibit the radical chain reaction [46]. The changes in peptide length and the amino acid content of peptides seen in protein hydrolysates may be responsible for the variations in their capacity to scavenge free radicals. Exploring bioactive peptides is another intriguing approach to creating meat products with useful properties. These useful substances, which result from microbial action, have antioxidant properties. A recent investigation of *L. plantarum* CD101 in a fermented sausage showed that short peptides derived from their inoculated starter were connected to increased antioxidant activity by DPPH radical-scavenging activity [21,23]. According to [21], it was found that peptides with the highest DPPH radical-scavenging activity and amino acid sequences obtained 44 peptides, for example E-G-A-D-S-E-M-A-L-F-G-E-A-A-P-Y-L-R-K-S-E-K-E-V-G-K-N and E-D-T-N-A-T-P-G-M-V-C-D-N-G-S-G-L-V-K from the meat muscle protein changes of myosin-1 and actin during proteolytic fermentation.

#### 3.4.2. FRAP-Reducing Power Capacity

The ferric reducing assay is regularly used to measure an antioxidant’s capacity to provide a free radical with an electron in the form of Fe^3+^ (ferricyanide complex) to Fe^2+^ (ferrous); the higher reducing power of each sample is better at donating electrons. In this assessment, the testing solution changes to a green and/or blue color, indicating that the higher reducing power of each hydrolysate gradually increases with fermentation time. Particularly, NPH-*nham*1 had the highest reducing power capacity of 73.7 mmole TROE/100 g sample at 10 mg/mL for 114 h, followed by the fraction of NPH-*nham*3 and NPH-*nham*2 inhibited at the 66.8 and 60.3 mmole TROE/100 g sample, respectively (Figure 3-right). This accords with our earlier observation, which showed that the highest FRAP-reducing power of the 70.1 µmole/mg fraction hydrolysate obtained the fraction of B_2_D_5_–3kDa-F_1_ purified from *plaa-som* fermentation with the additive pineapple enzymatic digestion [20]. A related study considered the antioxidant activity of sausages produced with a combination of *L. plantarum* CD101 and *S. simulans* NJ201 which results in a significant increase in the ferric reducing power and increases with the free radical scavenging of DPPH and ABTS activities [47]. Other research has reported that the peptide from sausage has the potential to inhibit the angiotensin-I-converting enzyme in a hypertensive and cardioprotective effect [17].

Additionally, much research has shown that peptides created from different fish protein hydrolysates function as potential antioxidants in terms of free radical-scavenging activity and reducing power capacity. As mentioned in the literature review, fish protein hydrolysates generates peptides using various enzyme sources, namely pepsin, trypsin, papain, alcalase, neutrase, pronase E, validase, and flavourzyme, which play a part in the proteolysis process. Some examples of these generated peptides were: sardine fish hydrolysate (L-Q-P-G-Q-G-Q-Q) [48], conger eel hydrolysate (L-G-L-N-G-D-D-V-N) [49], yellowfin sole protein hydrolysate (R-P-D-F-D-L-E-P-P-Y) [50], and pacific hake protein hydrolysate (P-L-F-Q-D-K-L-A-H-A-K and A-E-A-Q-K-Q-L-R) [51].

### 3.5. Antimicrobial Activity

The antibacterial activities of NPHs were evaluated against gram-negatives, such as *E. coli*, *Ent. aerogenes*, and *Sal. typhimurium*, beside the gram-positives of *S. aureus* bacteria. The antibacterial activity was assessed by evaluating the inhibition zone around the paper discs with a diameter of 7 mm. Interestingly, as can be seen in Table 4, NPH-*nham*1 exhibited a strong inhibitory activity against the four pathogenic bacteria of *E. coli*, *Ent. aerogenes*, *Sal. typhimurium*, and *S. aureus* in different clear zones of 22.0, 18.0, 17.0, and 16.0, respectively. In addition, NPH-*nham*2 and NPH-*nham*3 exerted significant inhibitory activity against all the indicator pathogens, while NPH-control showed inhibitory activity against only *Sal. typhimurium*. These findings are in accordance with other performance measurement systems that gram-negative bacteria are more resistant to antibacterial agents than gram-positive bacteria. In conclusion, NPH-*nham*1 and NPH-*nham*3 hydrolysates were discovered to have the most powerful antibacterial activity that is effective against the strain of *E. coli*, which is the main cause of food spoilage and also a cause of diarrhea in consumers. Additionally, the results are significant for *S. typhi*, *Sal. typhimurium*, *B. cereus*, and *S. aureus*, which are well known for producing a variety of enterotoxins that cause gastroenteritis and for being resistant to various phytochemical substances [52]. In general, fermented foods with antibacterial effects may be a result of either the peptides created during the breakdown of proteins or the bacteriocin produced by lactic acid bacteria. The antibacterial spectrum of fermented hydrolysates was therefore shown to be mostly caused by the peptides present rather than the strain utilized [18]. Most results in fish antimicrobial peptides have found antibacterial or bacteriostatic effects against a variety of gram-negative and gram-positive pathogens. The research of [22], who reported a novel antimicrobial peptide, ‘pelteobagrin’, isolated from the skin mucus of yellow catfish, shows bioactive peptides approximately 20 amino acids in length (G-K-L-N-L-F-L-S-R-L-E-I-L-K-L-F-V-G-A-L). The action of pelteobagrin displayed broad-spectrum activity against a variety of both pathogen microorganisms.

## 4. Conclusions

Overall, the current work demonstrated the in vitro probiotic test for resistance to the simulated human gastrointestinal tract and used high potential probiotics to ferment *Nham* (fermented pork sausage) through the encapsulation of probiotic starter cultures. This was verified in the in vitro part of the bioactive property test: through antioxidant and antimicrobial activities. Therefore, it can be concluded that the effect of using encapsulated probiotic starter cultures during the fermentation of *Nham* between physicochemical and microbiological characteristics significantly changes the protein muscle in NPHs hydrolysates for enhancing the potential bioactive peptide, mainly antioxidant and antimicrobial peptides. Based on these findings, the *Nham* probiotic of each formulation was fermented by encapsulating three different *Lactobacillus* starter cultures to generate NPHs hydrolysates. These hydrolysates have the potential and ability to be used in the food-processing industry as a natural preservative against food-borne infections and also to enhance the bioactive compounds for consumers’ health and well-being.

## Figures and Tables

**Figure 1 foods-11-03846-f001:**
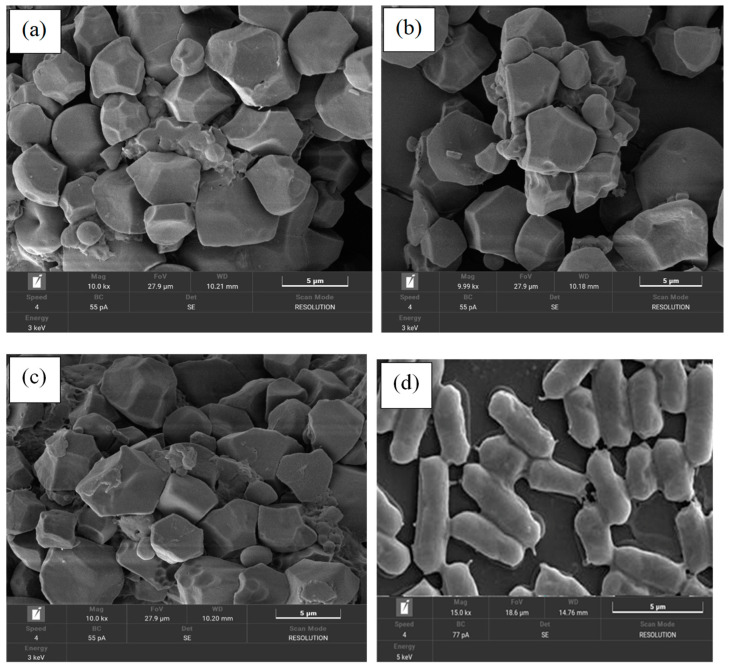
FE-SEM micrographs of the different encapsulations of the *Lactobacillus* species starter culture, with components of GRF combined with inulin at a ratio of 1:1 (% *w*/*v*): (**a**) encapsulation of *L. casei* KKU-KK1, (**b**) encapsulation of *L. pentosus* KKU-KK2, (**c**) encapsulation of *L. acidophilus* KKU-KK1, and (**d**) un-encapsulation of mixed *Lactobacillus* sp. cell culture, respectively.

**Figure 2 foods-11-03846-f002:**
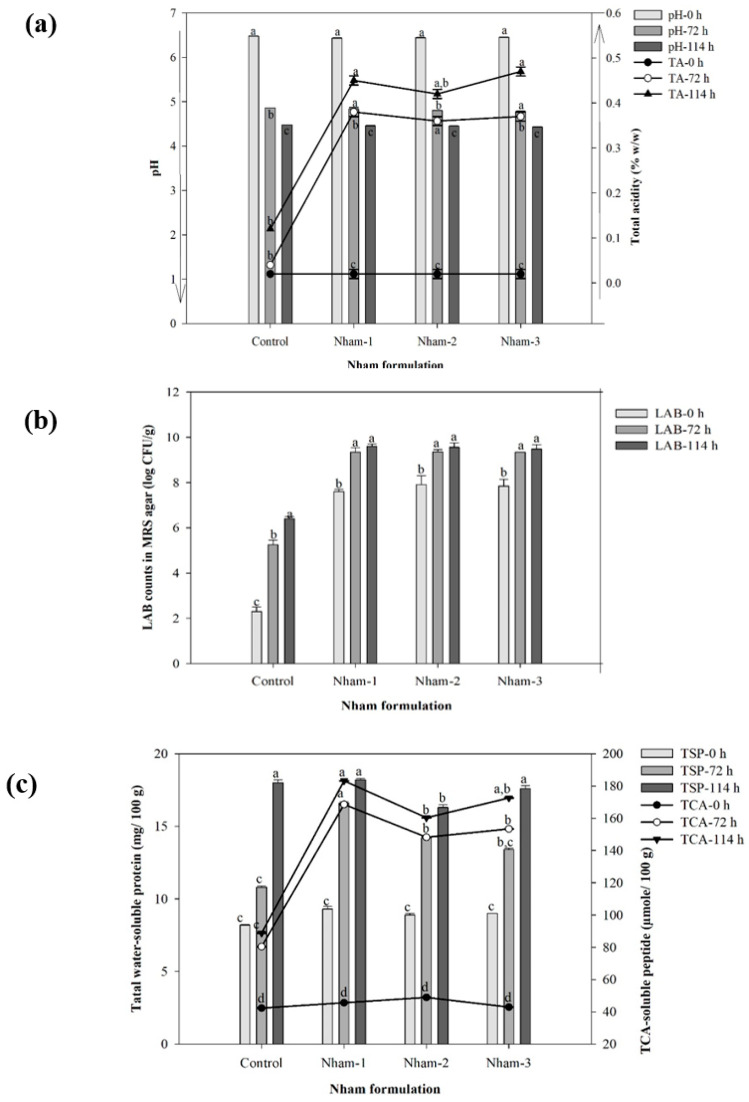
The co-relationship between physicochemical and microbiological characteristics of the changes in pH and TA (**a**), and (**b**) LAB counts during each *Nham* probiotic fermentation with different *Lactobacillus* sp. Mean values ± SD (*n* = 3) of different *Lactobacillus* sp. with different letters (**a**–**c**) for their properties during fermentation time.

**Figure 3 foods-11-03846-f003:**
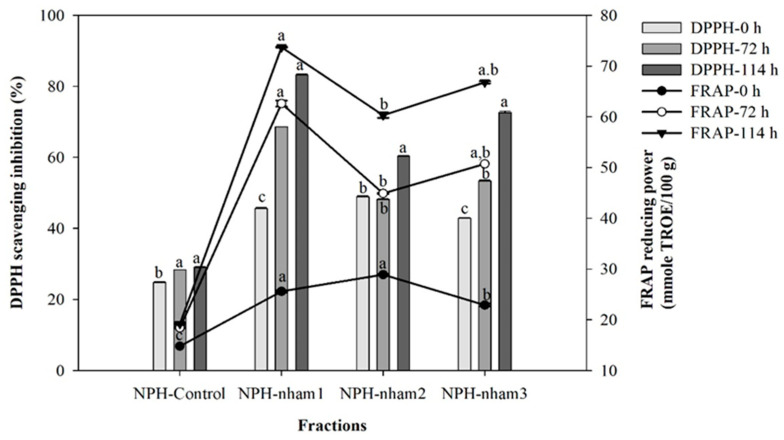
Antioxidant activity of different *Nham* protein hydrolysate fractions obtained during fermentation at different formulations, mixed with an encapsulated-*Lactobacillus* sp. starter. Different letters ^a,b,c^ in the same group indicate a significant difference by applying Duncan’s multiple range test (*p* < 0.05).

**Table 1 foods-11-03846-t001:** Formulation ratios of the probiotic starter encapsulation, consisting of *L. casei* KKU-KK1, *L. pentosus* KKU-KK2, and *L. acidophilus* KKU-KK3, for the *Nham* fermentation process.

Formulation	Mixing Ratio of Probiotic Starter Encapsulated Powder	Amount Ratio [% (*w*/*v*)]
Control	Spontaneous fermentation	no fill
*Nham*-1	*L. casei* KKU-KK1: *L. pentosus* KKU-KK2: *L. acidophilus* KKU-KK3	1:1:2
*Nham*-2	*L. casei* KKU-KK1: *L. pentosus* KKU-KK2: *L. acidophilus* KKU-KK3	2:1:2
*Nham*-3	*L. casei* KKU-KK1: *L. pentosus* KKU-KK2: *L. acidophilus* KKU-KK3	1:2:2

**Table 2 foods-11-03846-t002:** Probiotic characteristics, antimicrobial activities, and the percentage of survival rates of different *Lactobacillus* species of the before and after encapsulated probiotic starter cultures.

Probiotic Strains	(%) Survival Rate of Resistive Difference Conditions			Antimicrobial Activity (mm)		
	Lysozyme	pH (3.0)	Bile salt	*E. coli*	*S. aureus*	*Ent. aerogenes*
The probiotic potential before encapsulation						
KKU-KK1	76.6	85.3	75.2	17.0	19.8	19.4
KKU-KK2	70.2	65.0	60.2	9.7	12.3	18.0
KKU-KK3	75.8	65.2	68.0	16.2	11.5	7.7
The probiotic potential after encapsulation						
KKU-KK1	70.4	62.8	65.6	18.9	20.8	20.6
KKU-KK2	58.5	50.3	50.1	12.4	16.4	18.5
KKU-KK3	62.4	61.8	60.4	16.6	12.2	11.3

Values are expressed in percentage (%) for survival rates while reported in (mm) for antimicrobial activity. For each column of the survival rates of probiotic testing on the cell viability under different conditions (before and after encapsulation process).

**Table 3 foods-11-03846-t003:** The cell viability of the initial and later encapsulated probiotic starters and their encapsulation efficiency for maintaining cell survival using the freeze-drying technique.

Probiotic Strains	Enumeration of Cell Viability (log CFU/g)		Encapsulation Efficiency (%)	Size (µm)	Zeta (mV)
	Initial Cell Suspension	After Encapsulated			
KKU-KK1	12.5 ± 0.0 ^a^	12.5 ± 0.0 ^a^	100.0 ± 0.0 ^a^	10.8 ± 0.1 ^a^	−44.4 ± 0.3 ^a^
KKU-KK2	12.7 ± 0.0 ^a^	12.7 ± 0.0 ^a^	100.0 ± 0.0 ^a^	10.3 ± 0.3 ^a^	−43.8 ± 0.2 ^a^
KKU-KK3	12.8 ± 0.0 ^a^	12.8 ± 0.1 ^a^	100.0 ± 0.0 ^a^	10.5 ± 0.2 ^a^	−44.6 ± 0.0 ^a^

Values are expressed as the mean ± SD (*n* = 3). Each column gives the amount of cell viability at different stages, with the initial stage and later encapsulation stage related to the percentage of encapsulation efficiency with a different letter, ^a^, indicating a different significance (*p* < 0.05 by Duncan’s test).

**Table 4 foods-11-03846-t004:** Antimicrobial activity of *Nham* protein hydrolysates from different fractions at the highest antioxidant activity.

NPH	Inhibition Diameter Zone of Antimicrobial Activity (mm)			
	*E. coli*	*S. aureus*	*Ent. aerogenes*	*Sal. typhimurium*
NPH-control	8.0 ± 1.0 ^c^	8.0 ± 0.0 ^c^	7.0 ± 2.0 ^c^	-
NPH-*nham*1	22.0 ± 0.0 ^a^	16.0 ± 1.0 ^a^	18.0 ± 2.0 ^a^	17.0 ± 1.0 ^a^
NPH-*nham*2	16.0 ± 1.0 ^b^	11.0 ± 3.0 ^b^	14.0 ± 1.0 ^b^	15.0 ± 2.0 ^c^
NPH-*nham*3	20.0 ± 2.0 ^a,b^	13.0 ± 1.0 ^b^	15.0 ± 3.0 ^b^	16.0 ± 0.0 ^b^

Values are expressed as the mean ± SD (*n* = 3). Each column tests the antimicrobial activity with different letters, ^a, b, c^, indicating a different significance (*p* < 0.05 by Duncan’s test). NPHs as *Nham* protein hydrolysates; (-) as no inhibition.

## Data Availability

Data are contained within the article.

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
