# Peer review of "Bioaccessibility and Microencapsulation of Lactobacillus sp. to Enhance Nham Protein Hydrolysates in Thai Fermented Sausage"

_foods, 2022, doi:10.3390/foods11233846_

Round 1
Reviewer 1 Report
Congratulations, a well done study; however, I think it would be interesting to understand the reasons of using lactobacillus strains isolated from plaa som fermentation process, instead of using pure ATCC cultures
The manuscript presents interesting and important data on microencapsulation process of LBS, however, a few corrections are needed:
1. Abstract: The section containing on lines 32 to 38 needs to be rewritten. The word, “consequently” not need to be present. The following paragraph should state that the fractionation of Nham protein hydrolysates (NHPs) were prepared by microwave extraction process. The language on the abstract must be more specific and express the importance of Nham protein hydrolysates.
2. Materials and Methods: Why was it important to use the lactic acid bacteria sp. used for this study those obtained from isolates from the plaa-som fermentation process, instead of using pure ATCC cultures.
3. Results and discussion: The survival rate values of the probiotic presented on sections 261 to 263 are not equivalent to values presented on table 2; needs clarification. Table 2 shows that the survival rate under lysozyme, pH 3 and bile salt conditions is higher for the probiotic potentials before encapsulation. In addition, the use of letters, a,b,c to indicate the different conditions should be clarified.
Overall the study was conducted properly, appropriate and the best testing methods was employed.
Author Response
We would like to thank the reviewer for our manuscript's careful and thorough reading. The reviewer's comments and suggestions, which help improve this manuscript's quality, are appreciated. All issues indicated by the reviewer are addressed as follows attack file

Reviewer 2 Report
Dear Editors and authors,
1- The modern nomenclature of lactic acid bacteria has changed the names of many species and genera and has been adopted since 2020, but the authors still use the old nomenclature. The names of bacteria must be modified according to the modern nomenclature, for example. Lactobacillus casei correct to Lacticaseibacillus casei.
2-The aim of the study is unclear and written at length in length. The aim of the study must be rewritten in a clear and concise manner that matches the results of the study.
3-Some of the paragraphs in the introduction need to be supported by modern scientific references, See line 53-57, I suggest you to add (Niamah, A. K., Al-Sahlany, S. T. G., Ibrahim, S. A., Verma, D. K., Thakur, M., Singh, S., ... & Utama, G. L. (2021). Electro-hydrodynamic processing for encapsulation of probiotics: A review on recent trends, technological development, challenges and future prospect. Food Bioscience, 44, 101458.)
4-Some of the working methods need to be supported by scientific references, such as Preparation of culture suspensions method and Estimation of probiotic-encapsulation with probiotic property method, I suggest you to add
1-Zhao, M., Huang, X., Zhang, H., Zhang, Y., Gänzle, M., Yang, N., ... & Fang, Y. (2020). Probiotic encapsulation in water-in-water emulsion via heteroprotein complex coacervation of type-A gelatin/sodium caseinate. Food Hydrocolloids, 105, 105790.
2-Afzaal, M., Saeed, F., Ateeq, H., Imran, A., Yasmin, I., Shahid, A., ... & Awuchi, C. G. (2022). Survivability of probiotics under hostile conditions as affected by prebiotic-based encapsulating materials. International Journal of Food Properties, 25(1), 2044-2054.
4-Estimation of probiotic-encapsulation with probiotic property method is unclear, What are the sources of pathogenic bacteria? How did you get active? What was culture medium you tested? What is the number of added bacteria?
5-Antimicrobial activity method, line 166, What is the size of the bacterial inoculation added?
6-Antimicrobial activity of NPHs method, Some types of bacteria such as Salmonella typhimurium are not mentioned in the bacterial isolates see page 3 line 102-113 and section 2.4, but we find these bacteria in this method!!!
7-Table 1, It is preferable to write the name of the bacteria instead of close symbols or give the bacteria different symbols.
8-Page 13 line 502, Are the Discussion or Conclusions?,See line 245 Page 6.
Author Response
We would like to thank the reviewer for our manuscript's careful and thorough reading. The reviewer's comments and suggestions, which help improve this manuscript's quality, are appreciated. All issues indicated by the reviewer are addressed as follows attack file.
